# Intelligent Dynamic Identification Technique of Industrial Products in a Robotic Workplace

**DOI:** 10.3390/s21051797

**Published:** 2021-03-05

**Authors:** Ján Vachálek, Dana Šišmišová, Pavol Vašek, Jan Rybář, Juraj Slovák, Matej Šimovec

**Affiliations:** 1Faculty of Mechanical Engineering, Slovak University of Technology in Bratislava, Námestie Slobody 17, 812 31 Bratislava, Slovakia; jan.rybar@stuba.sk (J.R.); juraj.slovak@stuba.sk (J.S.); matej.simovec@stuba.sk (M.Š.); 2SOVA Digital a.s. Bojnická 3, 831 04 Bratislava, Slovakia; pavol.vasek@sova.sk

**Keywords:** production line, color sensor, uncertainties, control charts, machine learning

## Abstract

The article deals with aspects of identifying industrial products in motion based on their color. An automated robotic workplace with a conveyor belt, robot and an industrial color sensor is created for this purpose. Measured data are processed in a database and then statistically evaluated in form of type A standard uncertainty and type B standard uncertainty, in order to obtain combined standard uncertainties results. Based on the acquired data, control charts of RGB color components for identified products are created. Influence of product speed on the measuring process identification and process stability is monitored. In case of identification uncertainty i.e., measured values are outside the limits of control charts, the K-nearest neighbor machine learning algorithm is used. This algorithm, based on the Euclidean distances to the classified value, estimates its most accurate iteration. This results into the comprehensive system for identification of product moving on conveyor belt, where based on the data collection and statistical analysis using machine learning, industry usage reliability is demonstrated.

## 1. Introduction

Requirements for fast and accurate product identification and their measured parameters are currently increasing in industrial production environments [1,2] Intelligent solutions with a multidisciplinary approach are one possible solution. In our case we decided to combine and use available statistical mathematical methods together with database and computer solutions based on machine learning. The combination of these approaches, allows us to provide appropriate solutions for industry deployment with fast response and accuracy, which would otherwise be difficult to implement.

One of the new challenges in the industry is rapid detection and identification of moving products with various parameters. Nowadays, in the spirit of intelligent industry, there is a trend to abandon mass series production and switch to customized small series production runs according to [3,4,5,6,7,8]. We know many methods for products detection such as (the most commonly used) barcodes, quick response (QR), radio frequency identification (RFID) codes etc. [9]. From a production point of view, recognition price and speed are decisive factors. Therefore we have concentrated in this article on a low-cost universal industrial color sensor [10]. From a product recognition accuracy perspective, static product recognition is the best, i.e., to perform product identification while the conveyor belt is stopped. Although this procedure is the most accurate, it wastes time within the production cycle. Therefore, we deal with the dynamic recognition of products in motion. This is more complicated in terms of accuracy, because uncertainties arise in products’ identification, when the measured value is outside the control chart limits.

There are more color recognition methods based on color calibration algorithms which are suitable for our case. These depend on color models representing the respective color space such as RGB and its subset standard RGB (sRGB), or cyan magenta yellow and key (CMYK), or luminance (Y), blue–luminance (U), red–luminance (V) (YUV), hue saturation value (HSV), hue saturation brightness (HSB) and hue saturation lightness (HSL) or CIELAB color space, as stated in [11,12,13]. Since the selected industrial color sensor uses RGB color space directly, rather specific algorithms such as thin plate spline integration (TPS-3D), partial least squares analysis (PLS) or commercial calibration algorithms, e.g., ProfileMaker (PROM) would be more capable from our point of view [14]. However, to ensure the fastest possible implementation, we decided to apply the K-nearest neighbor machine learning algorithm described in [15,16], with which we already had experience and useful results from previous practical experiments.

## 2. Subject and Methods

The main objective of the article is to present a universal solution for fast detection of industrial products based on their color. In order to have sufficient base of relevant data, we have prepared the sample workplace shown in Figure 1a. This workplace fully replicates industrial applications and is equipped with industrial components such as a KUKA KR3R540 robot (KUKA Deutschland GmbH, Augsburg, Germany), a SICK CSM-WP117A2P color sensor (SICK AG, Waldkirch, Germany) and a conveyor belt from Automatica (Liptovský Mikuláš, Slovakia).

We performed 21,600 measurements in total to obtain statistical data to determine whether the accuracy of sensor is sufficient for moving product identification as shown in Figure 2.

We evaluated Type A standard uncertainty for each product (color) separately from these measurements. Then we calculated Type B standard uncertainties and subsequently the resulting combined uncertainty, based on which we declare the best settings for the simulated process and after for the real operation. The next step is to set regulatory limits and create control charts for accurate identification. We can then evaluate if identified products are within the control limits and their identification is unambiguous. In other case, they are outside the control limits and their identification is ambiguous [17]. For these cases, we use the machine learning algorithm K-nearest neighbors. We have used values acquired during the measurements of color sensor accuracy as training dataset.

The methodologies are explained and analyzed in more detail in the following subsections.

### Preparation of Test Robotic Workplace

It was necessary to set up workplace to test identification by the color sensor which simulates real operation and it is automated. An automated workplace enables one to perform a large number of measurements using different combinations of the monitored factors. The workplace is represented by a conveyor belt, along which colored calibration cubes move. After passing the cube and performing a measurement, the cube is caught by the robot’s vacuum gripper and moved again to the beginning of conveyor belt. This process is repeated for given number of measurements. The color sensor is located on the side of conveyor belt so that its detection zone faces the belt. To attach and position sensor, we designed a tool, which was printed on 3D printer. We first designed and assembled the workplace using the Process Simulate simulation tool (Siemens PLM Software, Plano, Texas, USA). We tested there the reachability of individual points needed for execution of catching and releasing cube operations. During testing the robot’s declared range in the Process Simulate tool, we found this range was insufficient for the application. Therefore it was increased by adding a flange designed to allow insertion of a suction cup mechanism. After repeated simulation of the flanged robot range, a flange prototype was created on a 3D printer. The robot range was also verified at the physical workplace, as shown in Figure 3.

The KUKA KR3 R540 robot is a low payload capacity robot. However, in our case it exceeds the experiment requirements, as the weight of the transferred calibration cubes is up to 100 g. The repeatability of return to programmed position for this robot reaches a value of 0.02 mm, which is sufficient for our purposes. We also took this information into account when calculating the resulting uncertainty of the color sensor measurements, according to [18,19]. The primary parameters of the KUKA KR3 R540 robot are listed in Table 1.

We installed the KUKA ethernet KUKA Robot Language (KRL) extension into the robot for measurement process purposes. This extension allowed us to communicate via an Ethernet connection between the robot and a computer and collect the data. After correct configuration, it was possible to monitor inputs coming to the robot and based on them control outputs, in our case the conveyor belt and suction cup.

We used a conveyor belt from Automatica in the workplace implementation. The conveyor belt specifications are shown in Table 2.

The motor speed was regulated to 30 percent of the maximum revolutions per minute (RPM), which corresponded to the setting in real operation. The frequency converter was controlled by the robot’s output signals based on data from the control computer.

We chose the CSM-WP117A2P color sensor from SICK shown in Figure 4a for color cubes identification,. The compact size of this sensor, which facilitates its placement at the workplace, is one of its advantages. The sensor emits white light on the scanned object using additive color mixing from three color diodes. Based on the reflection of light it evaluates combinations of red, green and blue components, which finally define the scanned color. Measurements are performed based on the light beam emitted by the sensor. Due to the small beam size, a large color area is not required for accurate color identification. The color sensor’s primary technical data are listed in Table 3. The sensor’s relative sensitivity curve shows a dependency on the sensing distance as seen in Figure 4b.

Another advantage is the possibility of using communication via an input output (IO) link. We integrated the sensor using the Sensor Integration Gateway (SIG100). The advantage of such an integration is ability to connect multiple sensors through a single gateway. This significantly facilitates communication and data collection from the sensors. The SIG100 uses the Representational State Transfer (REST) application programming interface, so it was possible to query data from the sensor via a Java Script Object Notation (JSON) string generated by the control computer. After obtaining data from the sensor, this data was recorded to the Structured Query Language (SQL) database MySQL, which made their categorization and evaluation easier.

The measured output of the CSM WP117A2P sensor is three numerical values. Each value represents a percentage of a primary color. Values range in size from 0 to 100 percent. The first value represents percentage of red (R), the second percentage of green (G), and the third value percentage of blue (B) of the subject color. Based on this information, we defined following measurement model according to [21,22]:(1)$$\delta_{Color}=\left[\delta_{R},\;\delta_{G},\delta_{B}\right]$$
where $\delta_{Color}$ is the resulting composite color percentage calculated from the contributions of the red $\delta_{R}$, green $\delta_{G}$ and blue component $\delta_{B}$. When determining the overall measurement result, we did not consider correlations of values due to the fact that this is new proposal for which it is necessary to perform further experiments.

## 3. Results

In our experimental measurements using the color sensor, we found that the highest influences on the measured values came from the illumination of the scanned object, the distance of the sensor to scanned object and whether the scanned object was moving or stopped [22,23,24]. Therefore we decided to perform measurements with different combinations of these factors. The optimum sensing distance specified by the manufacturer is 12.5 mm, with a tolerance of 3 mm. We chose values approaching to the limit distances of 15 mm, 10 mm and the optimal measuring distance of 12.5 mm. The illumination of the scanned object was another major influence on the measurements. We performed the measurements in natural daylight, with artificial light and in the dark. As a main goal of these measurements was to determine whether the measurement accuracy would be sufficient to identify the cubes even when the conveyor belt is running, therefore we performed measurements of stopped and moving cubes and compared the deviations.

We measured all settings combinations for six cubes with an edge size of 30 mm and following colors: red, blue, pink, green, yellow and brown. The measured calibration cubes are shown in Figure 1b.

By combining the influencing factors we created 18 combinations. We performed 200 measurements for each of the cube, from which we created the resulting dataset containing 21,600 measurements.

In the first phase we evaluated Type A standard uncertainty for individual color components (red—$u_{A_{R}}$, green—$u_{A_{G}}$, blue—$u_{A_{B}}$) measured by the sensor. Subsequently we calculated total Type A standard uncertainty ($u_{total}\;$) for each given combination. The following part of the work provides the results of our Type A standard uncertainty evaluation for individual colors for the abovementioned combinations, in a tabular form. When calculating uncertainties, we used procedures published in the literature [22,25,26].

Table 4 shows the Type A standard uncertainty calculated from measurements performed on a red cube. The table shows that smallest uncertainty was achieved when the cube is stopped under artificial light at a scanning distance of 10 mm. On the other hand, uncertainties calculated from measurements for a cube in motion on the conveyor are higher for all combinations of factors than for a stopped cube. This fact confirms that movement of the conveyor has a significant effect on measurement. The highest total uncertainty was achieved when measuring a cube in motion under artificial light at a sensing distance of 12.5 mm.

Table 5 shows resulting Type A standard uncertainty for a blue cube. The lowest uncertainty is shown for measurements performed when the cube is stopped in the dark with a distance of 12.5 mm between the sensor and the scanned object. The highest uncertainty was achieved when measuring the cube moving along the conveyor belt under artificial light at a distance of 10 mm.

Table 6 lists the Type A standard uncertainty calculated from data obtained when a pink cube was measured. The lowest total uncertainty was recorded for a static cube in the dark at the distance of 15 mm. The highest uncertainty was recorded when measuring the cube in motion under artificial light at a distance of 10 mm.

Table 7 shows the total Type A standard uncertainty calculated for green cube measurements. The lowest uncertainty was achieved when the cube is stopped under artificial light at a measuring distance of 10 mm. The highest uncertainty was recorded for measurements performed for a moving cube in natural daylight at a distance of 15 mm.

Table 8 lists Type A the standard uncertainty calculated for a yellow cube. The table shows that the lowest uncertainty was achieved when the cube is stopped under artificial light at a distance of 12.5 mm. The highest uncertainty was achieved when measuring a moving cube under artificial light at a distance of 10 mm.

Table 9 contains Type A standard uncertainty calculated for measurements performed for a brown cube. The lowest total uncertainty was recorded when the was stopped under natural daylight at a distance of 10 mm. The highest uncertainty was recorded when measuring a cube in motion in the dark at a distance of 10 mm.

When we compare all calibration cubes, the lowest Type A standard uncertainty shown in the dataset is for the brown cube, when a stopped cube is measured under different conditions. The highest uncertainty is shown by measurements performed for pink cubes. When evaluating the data measured on the conveyor without stopping the cube, measurements made on the green cube achieve the lowest Type A standard uncertainty, while values measured for the yellow cube show the highest uncertainty. By further examining data from the objects’ illumination point of view, we came to the conclusion that values of Type A uncertainties are lowest when measuring in the dark. This conclusion was confirmed by data obtained in both static and motion measurements. The highest Type A uncertainties were recorded in daylight static measurements, which may be due to its variance. When measuring in motion, we recorded the highest Type A uncertainties for artificial light measurements, which could be affected by reflection of light from moving objects.

From the objects’ distance point of view the lowest Type A uncertainties corresponded to the data measured at a distance of 15 mm and the highest uncertainties were for measurements performed at a distance of 10 mm between the sensor and the measured object. We came to the same findings when evaluating the data from static as well as motion measurements.

As already mentioned in the tables, measurements performed for cubes in motion showed several times higher Type A uncertainties for all colors and all factor combinations than measurements performed for static cubes.

In the following part of the article we focus on Type B uncertainties, which we determined based on identifiable sources of uncertainties affecting the measurements. After our analysis of the measurement process, we identified six components of Type B uncertainty. These sources and their value distributions are listed in Table 10 [22,25,26].

The first identified component of Type B uncertainty is the uncertainty of cube placement by the robot. We estimated this value of uncertainty based on the repeatability of the robot’s return into a specified position. This is stated in the robot documentation, according to standards for industrial manipulators. We also took into account the calibration deviation of the robot effector and flange shape deviation caused by inaccurate assembly of individual 3D printed parts. Based on the combination of these factors, we finally estimated the resulting component called cube placement by robot, hereinafter represented by $u_{B1}$.

As the second component of Type B uncertainty, we identified the sensitivity of the sensor at different sensing distances. Since we measured at three distances between the sensor and the scanned object, namely distances of 10, 12.5 and 15 mm, we determined its sensitivity at the mentioned distances based on sensor documentation. The sensitivity for the individual distances is hereinafter referred to as $u_{B2}$ and its values for individual distances are given in Table 10.

The third component of Type B uncertainty is the effect of illumination. We determined this component by estimation based on experimental measurements. As already mentioned in previous subsections, data measured in the dark achieved the lowest variability. Based on this, the lowest uncertainty value was assigned to this uncertainty component in the dark measurements. The uncertainty values for individual illumination are given in Table 10 with the designation $u_{B3}$.

As the next component of Type B uncertainty we determined the effect of conveyor movement. When estimating the value of this uncertainty component, we used the documentation of the conveyor belt motor, specifically the motor speed. The values of this uncertainty component for measurements of static cubes and moving cubes are given in Table 10 under the designation $u_{B4}$.

The component $u_{B5}$ was determined from the range of measured values, based on the difference between the maximum and minimum measured values of individual color components. We determined the interval for each combination of measurement settings for each color. The resulting interval of calculated values is shown in Table 10.

The last component of Type B uncertainty is the microclimate. This component includes the effect of the external environment on the measurements. The determined value of this uncertainty component is given in Table 10, defined according to [21].

Based on the uncertainty components listed in Table 10, we calculated the combined standard uncertainty and expanded uncertainty. When determining the expanded uncertainty, we chose the expansion coefficient k=2 i.e., the Gaussian distribution for a coverage probability of 95.45%.

We calculated the combined standard uncertainty based on the total standard Type A uncertainty from data in Table 4, Table 5, Table 6, Table 7, Table 8 and Table 9 and the total standard Type B uncertainty given in Table 10. These total uncertainties, listed as $u_{A_{total}}$ and $u_{B_{total}}$, are recalculated for every color cube, representing the customized product separately.

We can state based on these calculations (a sample calculation is presented in Section 3.1) that the influence of conveyor belt movement on the measurement uncertainties is considerable. Measurements made on a moving conveyor belt show higher values of uncertainties. This was confirmed for all cube colors and combinations of the measurement process settings.

During data evaluation we found the lowest uncertainty was recorded for measurements performed on brown cube. The brown cube achieved the best results when measuring on a stopped or even moving conveyor belt. On the other hand, the highest uncertainty was achieved for a static green cube and for a yellow cube in motion.

We investigated the effect of sensor distance from the scanned object as the next factor. The data indicate 15 mm as most suitable measuring distance for both static and motion measurements. In the static measurements, the highest uncertainties were recorded when measuring at a distance of 12.5 mm. Measurements in motion reached the highest uncertainties at a measuring distance of 10 mm. It can be stated that the measuring affects motion measurements more than static ones. While for static measurements the uncertainties at individual distances achieve similar values, for motion measurements the uncertainty at a distance of 10 mm is significantly higher than at distances of 12.5 and 15 mm.

When evaluating the uncertainties of the data in terms of the lighting used, we found the lowest measurement uncertainties for measurements of stopped as well as moving cubes on the conveyor, accomplished in the dark. We recorded the highest values of uncertainties when measuring under artificial light. For static measurements, the value of the uncertainties under artificial light is significantly higher than in daylight. This is not the case for measurements in motion, where the uncertainties recorded in daylight and under artificial light have very similar values, which may be caused by cube color reflections.

Our findings show that if we want to use all colors of cubes, the best combination of settings of the measurement process is a measuring distance of 15 mm and the measurement should be performed in the dark, ideally with stopped cubes.

### 3.1. Statistical Evaluation of Measured Data

To demonstrate a sample procedure of statistical evaluation of uncertainties, we chose a brown color cube as the customized product, based on the best results whether stopped or in motion. Table 11 shows values of individual measured color RGB components, measured for a moving brown cube at a scanning distance of 15 mm, in the dark. Column R indicates the percentage of red component, column G indicates the percentage of green component and column B indicates the percentage of blue component, respectively.

In the first evaluation step we determined the standard Type A uncertainty for individual color components using statistical methods, according to Equations (3) and (4).
(2)$$u_{A_{R}}=\sqrt{\frac{\sum_{i=1}^{n}{\left(x_{Ri}-{\bar{x}}_{R}\right)}^{2}}{n\left(n-1\right)}}=0.016\%$$
(3)$$u_{A_{G}}=\sqrt{\frac{\sum_{i=1}^{n}{\left(x_{Gi}-{\bar{x}}_{G}\right)}^{2}}{n\left(n-1\right)}}=0.006\%$$
(4)$$u_{A_{B}}=\sqrt{\frac{\sum_{i=1}^{n}{\left(x_{Bi}-{\bar{x}}_{B}\right)}^{2}}{n\left(n-1\right)}}=0.011\%$$

Total Type A standard uncertainty is then [11,15]:(5)$$u_{A_{total}}=\sqrt{u_{A_{R}}^{2}+u_{A_{G}}^{2}+u_{A_{B}}^{2}}=0.180\%$$

In the next step, we defined the individual sources of Type B uncertainties based on the equipment documentation and our experimental measurements. The sources of uncertainties we used for determination of combined standard uncertainty are shown in Table 12, as mentioned in [18].

The first uncertainty component is repeatability and it indicates the total standard uncertainty determined by the Type A method. The uncertainty component called cube placement by the robot determined by the type B method, was estimated based on the return repeatability of the robot manipulator with the considered assembly deviation of the robot vacuum gripper. The uncertainty component sensor distance sensitivity was determined based on the documentation of the CSM-WP117A2P color sensor. When determining the illumination effect, we used experimental data obtained with different types of lighting (darkness, daylight, artificial light) from experiments. The uncertainty component called conveyor movement effect is calculated based on the conveyor speed specified in its documentation. The uncertainty component range of measured values was determined based on calculations performed on our experimental data. The influence of microclimate includes the impact of the surrounding environment, in our case an air-conditioned laboratory as mentioned in [19,25].

From the individual components of uncertainty determined by the type B method, we subsequently calculated a total Type B uncertainty according to the following formula:(6)$$u_{B_{total}}=\sqrt{u_{B_{1}}^{2}+u_{B_{2}}^{2}+u_{B_{3}}^{2}+u_{B_{4}}^{2}+u_{B_{5}}^{2}+u_{B_{6}}^{2}}=1.410\%$$

After calculation of the total Type A standard uncertainty and total Type B standard uncertainty, we determined the combined standard uncertainty based on a formula given in [18]:(7)$$u_{C}=\sqrt{u_{A_{total}}^{2}+u_{B_{total}}^{2}}=1.422\%$$

When defining the expanded uncertainty, we chose the expansion coefficient k=2 i.e., the Gaussian distribution for a coverage probability of 95.45%. The relationship in the sense of [19,26] then applies to the expanded uncertainty:(8)$$U=k\cdot u_{C}=2.844\%$$

The result of color measurements (for the brown calibration cube) using the CSM-WP117A2P color sensor after merging the color components and rounding, according to the balance table in Table 13, can be written as follows:

The resulting color =(82.857±2.844)%;k=2.

Similar calculations were applied to all the other colors representing customized industrial products. Based on the results, the brown color was evaluated as the best and therefore its sample calculation was presented in this subsection.

### 3.2. Definition of Control Limits and Control Chart Creation

For proper functionality of a logistics system, it is necessary to ensure that sensors located at workplaces are able to correctly identify calibration cubes based on red, green and blue color components. We decided to use the control chart statistical tool to ensure the stability of the measurement process. This is a graphical method using the principle of statistical tests of significance. The purpose of control charts is to compare and visualize the current state of measurement with respect to predefined limits. When defining the limits, we take into account the internal variability of the measured process. The basic parts of a Shewart control chart are [27,28]: central line (CL); upper control limit (UCL); and lower control limit (LCL).

The central line represents the reference value of the displayed characteristic, in our case the average of measured values $\bar{X}$. We assume a normal distribution therefore the control limits are set at the distance of 3σ on both sides of the central line, where σ denotes the standard deviation. The control limit above the central line represents the high control limit and the control limit below the central line represents the low control limit.

We initially assumed the possibility of creating two charts for each calibration cube, which would cover all combinations of the measurement process settings. While one of the charts would monitor the process stability when measuring stopped cubes, the second would do this for a moving conveyor belt. As mentioned in the evaluation of uncertainties, data measured during the movement of cubes on the conveyor belt showed several times higher uncertainties than data measured when the calibration cube was stopped. Based on this finding, two control charts would allow us to define control limits for a larger number of colors in static measurements without overlapping.

The color component values of the measured color within one setting combination, show a relatively low degree of variability. However, comparison of individual sets showed that the average measured values of color components differs depending on the settings combination used. Further testing revealed that illumination is the main factor influencing the change of these values. Other variability of values occurred for scanning distance changes, but this difference was not so significant. Based on these findings, the best way to increase the number of identifiable colors would be to create a database of control limits for each combination of measurement process settings. Limits could be then selected based on the particular conditions at the measurement site.

From the part of this work describing measurement uncertainties, we can see the lowest achieved measurement uncertainties correspond to the brown calibration cube. The best combination of measurement settings was in the dark, at a distance of 15 mm. Therefore, we chose a brown cube measured in the dark, at a scanning distance of 15 mm, as an example for determining the stability of the measuring process using control charts.

### 3.3. RGB Color Component Charts for Measurement of Stopped Brown Cube

Figure 5, Figure 6 and Figure 7 show the color components control charts created from measurements of a brown cube in the dark at a scanning distance of 15 mm. We used the whole set of measurements for a given settings combination when determining the control limits, but we printed only the first 45 measured values to maintain the clarity of the charts. The vertical axis of the charts shows the values of the measured color components in percentages. The horizontal axis shows the measurement number at which the value was recorded. We used procedures described in the literature to define the control limits [29].

Figure 5 show red color components control chart. When determining the control limits, we started with a normal distribution i.e., we set the control limits at a distance of 3σ from the central line. However, in the case of a stopped cube, this distance was not sufficient for the upper control limit, as several values exceeded this limit, which caused instability of the control chart. Therefore we moved the upper control limit to the distance of 4σ, which proved to be sufficient to ensure the stability of the control chart based on the visualized dataset, as all measured values were within control limits.

Similarly we set the limits for the green and blue color components. After application of the measured data, we found the that control charts are stable. The control chart for the green color component is shown in Figure 6 and the control chart for the blue color component is shown in Figure 7. As we can see from the charts, the lowest variability in values was identified for the green color component. We set limits based on the normal distribution, and in the case of the green color component, these were sufficient to ensure the stability of the control chart and it was not necessary to expand them. In the case of the blue color component for a stopped cube, we extended the lower control limit to 4σ. This covered all measured data, and also did not unnecessarily expand the interval too much by shifting both control limits.

### 3.4. Stability Monitoring of the Measuring Process

Based on the control charts created for the individual color components for all combinations of measurement settings for the calibration cubes, it is possible to continuously monitor the stability of the measurement process at individual workplaces in production. After implementation of a color sensor into the production process, a combination of settings is selected based on the measurement conditions at the given workplace and respective control charts are selected for the individual color components of calibration cubes. If the measuring process is stable i.e., all measured values are inside the control limits range, it will also ensure the smooth operation of the logistics system, because it will work with correct data.

Color identification is essential to ensure smooth logistics operations. Without correct color identification, inaccuracies in calculations of the current state of materials on the line arise. Inaccuracy has a negative impact on the functionality of the system in long running operation. By implementing control charts, we can identify values that are outside the control limits and examine their cause.

The aim of the application of control charts is to maintain the stability of the measuring process. In the case of process instability, it is necessary to assess each measured value individually. The value can be also excluded, if this is an isolated case with a large deviation from the control limits. If this case is repeated and values accumulate outside the control limits, it is possible to implement some of appropriate corrective mechanism, e.g., shortening the calibration interval, extending the control limits, reduction of defects on the calibration cube or sensor.

If one of the sensor readings writes to the database a value that is outside the control limits for that color component of the measured color, the color is not successfully identified. In the next part, we address the failure of color identification and the possibility to solve this issue with minimum impact on the logistics process. At the same time we must be able to identify the origin of measurement errors.

### 3.5. K-Nearest Neighbors Algorithm for Identification of Values Outside the Control Limits

After a value is scanned by a sensor, this value is recorded into the database table corresponding to a particular sensor, based on some unique sensor identifier. Measurement number which is unique in the table is assigned to the value. The measured values of individual color components are thereafter loaded into the logistics system. Once values are available, it is checked whether color components are within the control limits of any of the defined colors. The color is identified, if component values are inside the control limits. If values fall outside the control limits, we still need to identify the color to avoid the disruption of logistics processes, which is controlled by the colors. For identification of colors based on values that are outside the control limits, we decided to use the K-nearest neighbor machine learning algorithm, which we implemented in the Python programming language.

The K-nearest neighbor is classification algorithm often used in the analysis of large datasets based on common attributes. In the first step, the algorithm assigns training data to a certain group based on their designation. The training data in our case are values measured during our experimental measurements. These data have six independent variables based on which the resulting dependent variable is defined and determining the group [30]:Conveyor belt speed;Illumination;Scanning distance;Measured value of red, blue and green color component.

Since we divided the data for control charts according to combinations of measurement settings, the dependent variables conveyor belt speed, illumination and scanning distance are constant for all measured colors. Based on this fact, we need only three variables to define the dependent variable, in our case color. These variables are measured values of individual RGB color components.

The training sets for algorithm at specific setting of measurement parameters thus contain 1200 measurements i.e., 200 for each measured calibration cube. We divided the data into training and testing data, using a ratio of 80/20 (training/testing) for functionality testing purposes. We can test in that manner whether the algorithm has not adapted too much to the training data, and still be able to respond to a new dataset. As sample data set, we chose measurements performed on a moving conveyor belt, in the dark, at a scanning distance of 15 mm. Figure 8 shows distribution of training data based on the dependent variable, in our case color, where the coordinates of individual points are determined based on RGB coordinates. As we can see in Figure 8, the data are grouped according to the color of cubes and there are visible gaps between the individual colors. After application of the algorithm on the training data, we verified its accuracy on the test dataset. Thanks to the mentioned gaps, the algorithm achieved an accuracy of 100% in categorization of the test data. This algorithm configuration is subsequently used to check measured values outside the control limits.

If a measured value outside the control limits of any color occurs during the process, this value is tested by an algorithm trained for that setting. The algorithm attempts to classify this value. The K-nearest neighbor algorithm for classification uses the Euclidean distances of trained data to the value to be classified. According to the input parameter of algorithm, which is number of searched neighbors, it determines given number of points with the lowest possible Euclidean distance to the value to be classified. The class of the new value is identified based on the class where most of the selected neighbors belong to [15,31].

The trained algorithm is therefore a tool which can be used for immediate estimation of measured colors when recorded a value is outside the control limits. We can reduce inaccuracies between the real and digital control system by application of the algorithm and ensure the smooth process running. It is necessary in our case to set the maximum distance of the nearest neighbor, what can filter out the category of values created by incorrect measurements. If we did not set the maximum distance tolerance, each measurement would be identified as one of the colors, no matter how far the measured values were from the control limits of defined colors. The application of the algorithm is important, especially when deploying sensors in a new workplace, until all environmental influences are identified. However, the algorithm is only offers the possibility of temporarily identifying solutions for values outside the control limits. If large number of values are outside the control limits during the measurements, one of the abovementioned corrective mechanism must be applied e.g., an extension of the control limits.

## 4. Discussion

One of the main trends nowadays is a shift away from mass series production and a transition to custom production based on the increasing requirements of demanding customers [8,32]. The consequence of this trend is often several variations of the same product on one production line at the same time. Each variation has its own specifics which must be taken into account within the production process. For this reason, it is currently essential to be able to recognize products with the highest possible accuracy and speed. Every single stop due to products’ identification, increases the work cycle and prolongs the production time of a product. Technology with intelligent recognition capability is relatively expensive and difficult to maintain [33].

The article offers us an advantageous alternative to expensive and complex technology for dynamic scanning and identification of products in motion, in the form of using a cheap static industrial color sensor with simple maintenance and adjustment wherever the type of production allows. It involves a simple color sensor commonly used in the industry [34,35,36,37]. The difference here is its usage in dynamic identification, where it is not primarily suitable due to the sensor characteristics. Based on our research and experiments, we can responsibly declare that it is possible to use a simple sensor for dynamic (more complex) product identification. However, statistical methods must be used in order to obtain combined standard uncertainties and control charts for a given sensor [38,39]. Subsequently the mentioned K-nearest neighbor machine learning algorithm allows us to rectify any sensor errors in dynamic color scanning [16]. The static color sensor with the best scanning results of static products, by using proposed methodology described in the article, becomes capable of identifying products even in motion. On-the-fly identification speeds up the entire production line and thus allows us to produce more products in the same time. Not least of all, since the conveyor belt does not stop, it significantly extends the technology operating life (by minimizing the occurrence of mechanical shocks), reduces required service (because of less mechanical component damage) and saves electricity (thanks to skipped energy-inefficient start-ups).

We currently see a demand for product identification technology in industrial enterprises [30]. Therefore, we would like to focus on enhancement of identification methods in the future. We plan to incorporate of optical methods for dynamic product recognition using special 3D sensors which are already at our disposal. These include the Photoneo PhoXi [40] or cheaper camera OpenCV alternatives [41], where we anticipate their integration with standard Cognex industrial cameras, supported by additional software for intelligent product recognition [42].

## Figures and Tables

**Figure 1 sensors-21-01797-f001:**
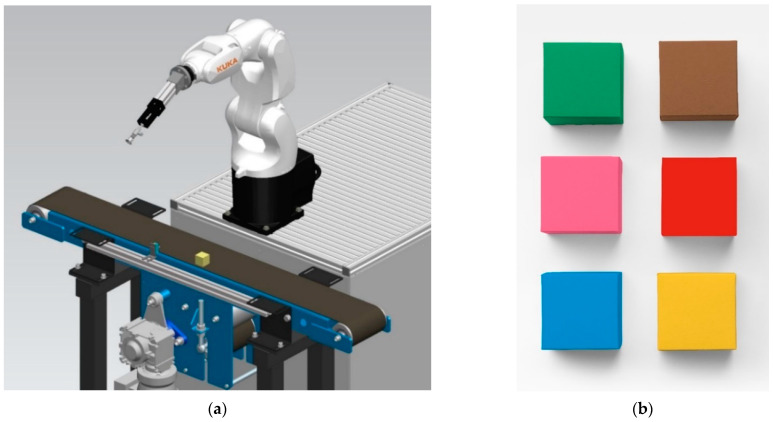
(**a**) Design of test robotic workplace in the Siemens Tecnomatix Process Simulate environment; (**b**) Calibration cubes for simulation of six different customized products.

**Figure 2 sensors-21-01797-f002:**
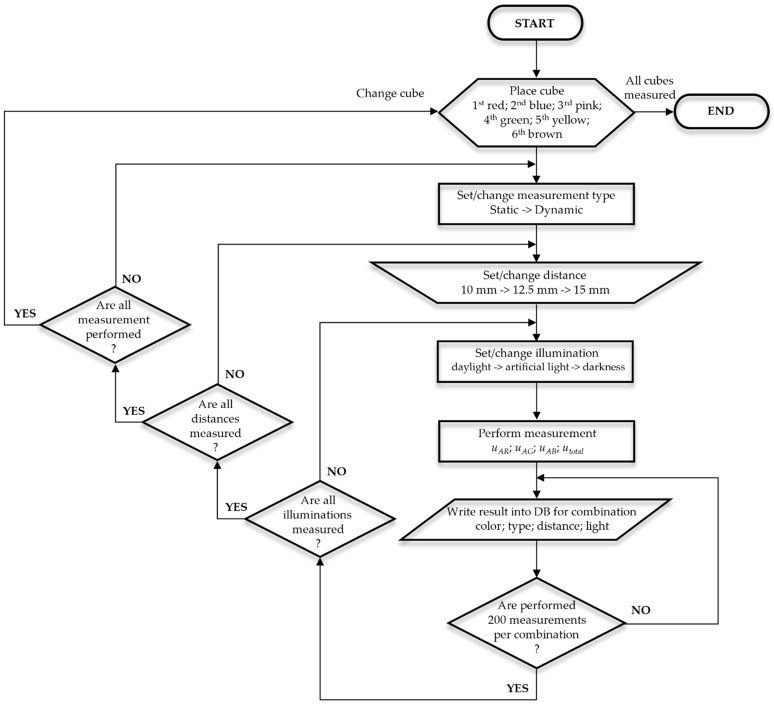
Measurement process workflow diagram.

**Figure 3 sensors-21-01797-f003:**
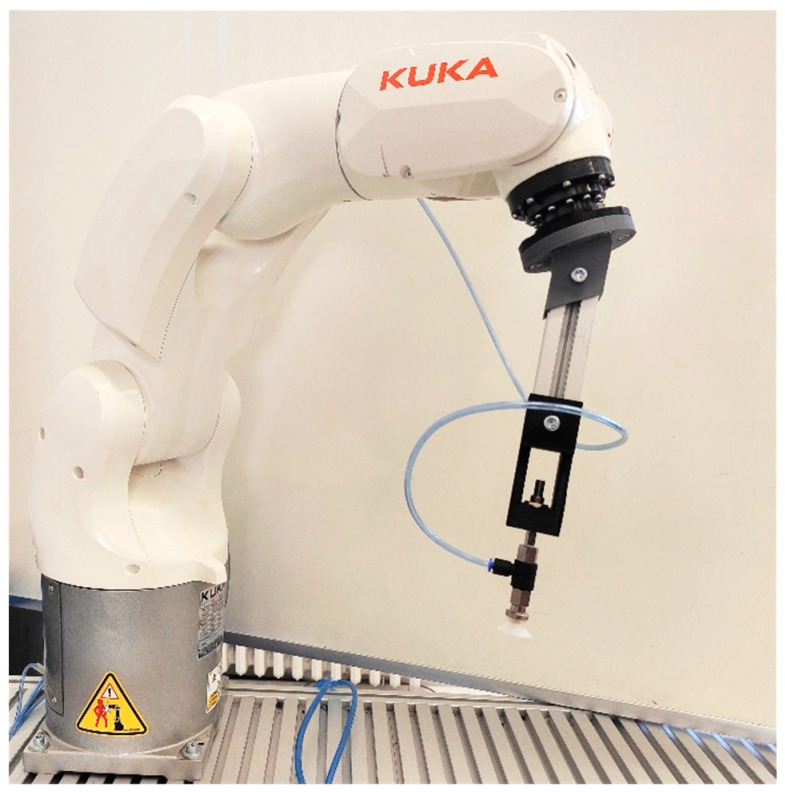
Robot range extension using flange.

**Figure 4 sensors-21-01797-f004:**
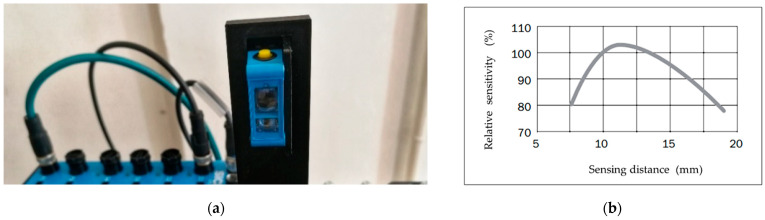
(**a**) Color sensor CSM—WP117A2P; (**b**) Sensing distance of sensor [20].

**Figure 5 sensors-21-01797-f005:**
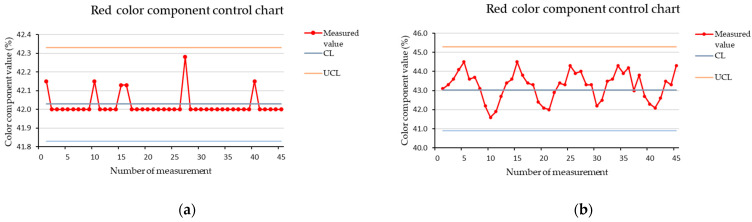
Red color component control chart for a brown cube measured in the dark, at a scanning distance of 15 mm, (**a**) stopped cube; (**b**) cube in motion.

**Figure 6 sensors-21-01797-f006:**
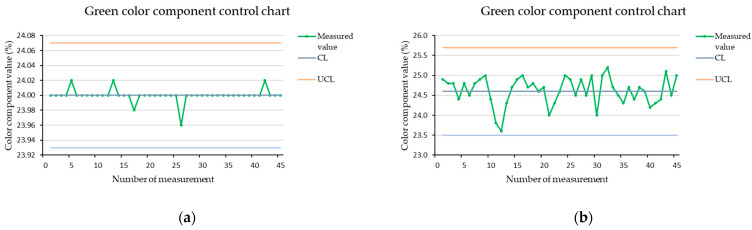
Green color component control chart for a brown cube measured in the dark, at a scanning distance of 15 mm; (**a**) stopped cube; (**b**) cube in motion.

**Figure 7 sensors-21-01797-f007:**
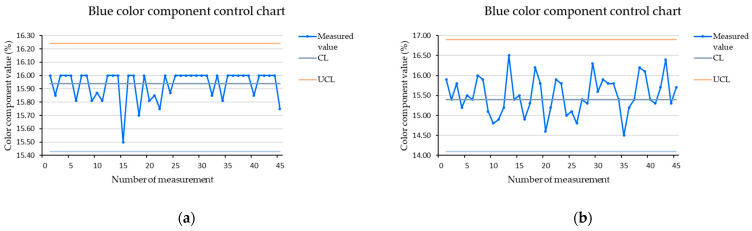
Blue color component control chart for a brown cube measured in the dark, at a scanning distance of 15 mm; (**a**) stopped cube; (**b**) cube in motion.

**Figure 8 sensors-21-01797-f008:**
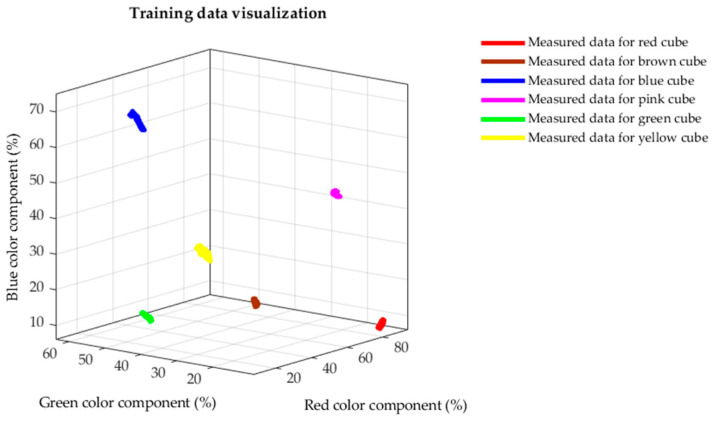
Visualized training data.

**Table 1 sensors-21-01797-t001:** KUKA KR3 R540 parameter overview.

Maximum reach	541 mm
Payload	3 kg
Pose repeatability	±0.02 mm
Number of axes	6
Mounting positions	Ceiling, Floor, Wall
Footprint	179 × 179 mm
Weigh	26.5 kg
Ambient operatingtemperature	5 °C–45 °C
Protection class	IP40
Controller	KR C-4 compact

**Table 2 sensors-21-01797-t002:** Automatica conveyor belt specifications.

Length	1500 mm
Width	25 mm
Height	1000 mm
Motor	three-phase motor Nord
Control	Siemens Sinamics V20 frequency converter
Maximum revolutions	1415 RPM
Belt material	rubber with anti-slip surface

**Table 3 sensors-21-01797-t003:** CSM-WP117A2P color sensor features.

Dimensions	22 mm × 12 mm × 32 mm
Sensing distance	12.5 mm
Sensing distance tolerance	±3 mm
Light source	Light Emitting Diode (LED), RGB
Wave length	640 nm, 525 nm, 470 nm
Light spot size	1.5 mm × 6.5 mm
Response time	300 µs
Supply voltage	DC 12 V ... 24 V
Output (channel)	1 color/8 colors via IO-Link

**Table 4 sensors-21-01797-t004:** Type A standard uncertainty for a red cube.

Illumination	Sensor Distance (mm)	Cube Stops				Cube in Motion			
		$u_{A_{R}}\left(\%\right)\;$	$u_{A_{G}}\left(\%\right)$	$u_{A_{B}}\left(\%\right)$	$u_{total}\left(\%\right)$	$u_{A_{R}}\left(\%\right)\;$	$u_{A_{G}}\left(\%\right)$	$u_{A_{B}}\left(\%\right)$	$u_{total}\left(\%\right)$
Natural daylight	10	0.010	0.005	0.011	0.016	0.105	0.021	0.026	0.110
Natural daylight	12.5	0.001	0.007	0.011	0.013	0.079	0.031	0.034	0.091
Natural daylight	15	0.004	0.011	0.001	0.012	0.060	0.023	0.020	0.067
Artificial light	10	0.000	0.003	0.001	0.003	0.198	0.036	0.043	0.206
Artificial light	12.5	0.002	0.003	0.001	0.004	0.458	0.041	0.051	0.463
Artificial light	15	0.004	0.001	0.011	0.012	0.065	0.030	0.028	0.077
Darkness	10	0.000	0.011	0.000	0.011	0.160	0.029	0.034	0.166
Darkness	12.5	0.016	0.000	0.008	0.018	0.064	0.025	0.027	0.074
Darkness	15	0.000	0.003	0.005	0.006	0.103	0.029	0.027	0.110

**Table 5 sensors-21-01797-t005:** Type A standard uncertainty for a blue cube.

Illumination	Sensor Distance (mm)	Cube Stops				Cube in Motion			
		$u_{A_{R}}\left(\%\right)\;$	$u_{A_{G}}\left(\%\right)$	$u_{A_{B}}\left(\%\right)$	$u_{total}\left(\%\right)$	$u_{A_{R}}\left(\%\right)\;$	$u_{A_{G}}\left(\%\right)$	$u_{A_{B}}\left(\%\right)$	$u_{total}\left(\%\right)$
Natural daylight	10	0.000	0.004	0.022	0.022	0.021	0.093	0.259	0.276
Natural daylight	12.5	0.011	0.003	0.004	0.012	0.020	0.027	0.049	0.059
Natural daylight	15	0.002	0.005	0.012	0.013	0.030	0.059	0.057	0.087
Artificial light	10	0.010	0.012	0.000	0.016	0.018	0.115	0.318	0.339
Artificial light	12.5	0.002	0.006	0.008	0.010	0.018	0.018	0.056	0.062
Artificial light	15	0.014	0.000	0.007	0.016	0.052	0.106	0.106	0.159
Darkness	10	0.014	0.001	0.002	0.014	0.022	0.095	0.235	0.254
Darkness	12.5	0.001	0.007	0.004	0.008	0.024	0.030	0.027	0.047
Darkness	15	0.011	0.003	0.007	0.013	0.042	0.111	0.113	0.164

**Table 6 sensors-21-01797-t006:** Type A standard uncertainty for a pink cube.

Illumination	Sensor Distance (mm)	Cube Stops				Cube in Motion			
		$u_{A_{R}}\left(\%\right)\;$	$u_{A_{G}}\left(\%\right)$	$u_{A_{B}}\left(\%\right)$	$u_{total}\left(\%\right)$	$u_{A_{R}}\left(\%\right)\;$	$u_{A_{G}}\left(\%\right)$	$u_{A_{B}}\left(\%\right)$	$u_{total}\left(\%\right)$
Natural daylight	10	0.003	0.005	0.000	0.006	0.119	0.035	0.066	0.141
Natural daylight	12.5	0.005	0.008	0.019	0.021	0.197	0.043	0.137	0.244
Natural daylight	15	0.006	0.003	0.016	0.017	0.035	0.042	0.037	0.066
Artificial light	10	0.018	0.022	0.010	0.030	0.192	0.218	0.230	0.371
Artificial light	12.5	0.000	0.015	0.019	0.024	0.213	0.084	0.131	0.264
Artificial light	15	0.006	0.007	0.012	0.015	0.077	0.053	0.036	0.100
Darkness	10	0.027	0.006	0.016	0.032	0.087	0.039	0.056	0.111
Darkness	12.5	0.000	0.014	0.014	0.020	0.149	0.028	0.074	0.169
Darkness	15	0.000	0.000	0.008	0.008	0.045	0.031	0.023	0.059

**Table 7 sensors-21-01797-t007:** Type A standard uncertainty for a green cube.

Illumination	Sensor Distance (mm)	Cube Stops				Cube in Motion			
		$u_{A_{R}}\left(\%\right)\;$	$u_{A_{G}}\left(\%\right)$	$u_{A_{B}}\left(\%\right)$	$u_{total}\left(\%\right)$	$u_{A_{R}}\left(\%\right)\;$	$u_{A_{G}}\left(\%\right)$	$u_{A_{B}}\left(\%\right)$	$u_{total}\left(\%\right)$
Natural daylight	10	0.007	0.012	0.001	0.014	0.028	0.093	0.048	0.108
Natural daylight	12.5	0.011	0.002	0.017	0.020	0.021	0.031	0.020	0.042
Natural daylight	15	0.004	0.014	0.004	0.015	0.053	0.179	0.041	0.191
Artificial light	10	0.005	0.002	0.002	0.006	0.044	0.147	0.079	0.173
Artificial light	12.5	0.014	0.019	0.012	0.026	0.023	0.034	0.027	0.049
Artificial light	15	0.001	0.016	0.012	0.020	0.026	0.045	0.024	0.057
Darkness	10	0.007	0.001	0.003	0.008	0.020	0.051	0.027	0.061
Darkness	12.5	0.000	0.004	0.008	0.009	0.020	0.029	0.020	0.041
Darkness	15	0.008	0.009	0.013	0.018	0.038	0.054	0.022	0.070

**Table 8 sensors-21-01797-t008:** Type A standard uncertainty for a yellow cube.

Illumination	Sensor Distance (mm)	Cube Stops				Cube in Motion			
		$u_{A_{R}}\left(\%\right)\;$	$u_{A_{G}}\left(\%\right)$	$u_{A_{B}}\left(\%\right)$	$u_{total}\left(\%\right)$	$u_{A_{R}}\left(\%\right)\;$	$u_{A_{G}}\left(\%\right)$	$u_{A_{B}}\left(\%\right)$	$u_{total}\left(\%\right)$
Natural daylight	10	0.001	0.000	0.013	0.013	0.292	0.280	0.125	0.423
Natural daylight	12.5	0.008	0.002	0.014	0.016	0.145	0.123	0.103	0.216
Natural daylight	15	0.014	0.001	0.004	0.015	0.172	0.144	0.048	0.229
Artificial light	10	0.019	0.003	0.003	0.019	0.383	0.333	0.112	0.520
Artificial light	12.5	0.004	0.005	0.000	0.006	0.118	0.104	0.092	0.182
Artificial light	15	0.014	0.007	0.011	0.019	0.144	0.112	0.027	0.184
Darkness	10	0.001	0.008	0.015	0.017	0.268	0.255	0.089	0.380
Darkness	12.5	0.002	0.014	0.010	0.017	0.158	0.099	0.078	0.202
Darkness	15	0.006	0.014	0.000	0.015	0.163	0.141	0.041	0.219

**Table 9 sensors-21-01797-t009:** Type A standard uncertainty for a brown cube.

Illumination	Sensor Distance (mm)	Cube Stops				Cube in Motion			
		$u_{A_{R}}\left(\%\right)\;$	$u_{A_{G}}\left(\%\right)$	$u_{A_{B}}\left(\%\right)$	$u_{total}\left(\%\right)$	$u_{A_{R}}\left(\%\right)\;$	$u_{A_{G}}\left(\%\right)$	$u_{A_{B}}\left(\%\right)$	$u_{total}\left(\%\right)$
Natural daylight	10	0.000	0.002	0.001	0.002	0.078	0.091	0.057	0.133
Natural daylight	12.5	0.008	0.008	0.014	0.018	0.043	0.056	0.051	0.087
Natural daylight	15	0.001	0.002	0.003	0.004	0.062	0.032	0.029	0.076
Artificial light	10	0.000	0.001	0.005	0.005	0.038	0.045	0.037	0.070
Artificial light	12.5	0.008	0.009	0.002	0.012	0.019	0.031	0.067	0.076
Artificial light	15	0.001	0.000	0.003	0.003	0.095	0.055	0.052	0.121
Darkness	10	0.000	0.006	0.001	0.006	0.117	0.116	0.085	0.185
Darkness	12.5	0.012	0.001	0.013	0.018	0.031	0.031	0.039	0.059
Darkness	15	0.005	0.000	0.007	0.009	0.050	0.026	0.033	0.065

**Table 10 sensors-21-01797-t010:** Type B uncertainty components.

Uncertainty Component	Uncertainty Type	Uncertainty Value	Distribution
Repeatability	$u_{A}$	In Table 4, Table 5, Table 6, Table 7, Table 8 and Table 9	---
Cube placement by the robot *	$u_{B1}$	0.02%	equal
Sensor distance sensitivity	$u_{B2}$	$u_{B2_{10mm}}=0.100\%$	equal
		$u_{B2_{12.5mm}}=0.105\%$	
		$u_{B2_{15mm}}=0.096\%$	
Illumination effect **	$u_{B3}$	$u_{B3_{darkness}}=0.700\%$	equal
		$u_{B3_{artificial}}=1.000\%$	
		$u_{B3_{daylight}}=0.800\%$	
Conveyor movement effect *	$u_{B4}$	$u_{B4_{motion}}=0.005\%$	equal
		$u_{B4_{static}}=0.000\%$	
Range of measured values	$u_{B5}$	(0÷7)%	equal
Microclimate ***	$u_{B6}$	0.1%	equal

* value estimated based on the documentation; ** value estimated from experimental measurements; *** estimated value.

**Table 11 sensors-21-01797-t011:** Type B uncertainty components.

Measurement Number	Red(%)	Green(%)	Blue(%)
1	43.433	24.600	15.743
2	42.886	24.457	15.267
3	43.100	24.457	15.800
4	42.200	24.378	14.767
5	41.725	23.933	15.267
6	42.767	24.800	15.600
7	43.100	24.350	15.475
8	43.400	24.725	15.600
9	44.225	24.600	14.767
10	43.850	24.600	14.600
11	43.433	24.711	14.800
12	43.100	24.711	15.171
13	43.000	24.933	15.933
14	42.100	23.933	15.433
15	42.600	24.725	15.800
∑	644.919	367.913	230.023
Average color representation	42.9946	24.52753	15.33487
The resulting color	82.857		

**Table 12 sensors-21-01797-t012:** Uncertainty sources for the brown cube.

Uncertainty Component	Uncertainty Type	Uncertainty Value	Distribution
Repeatability	$u_{A}$	0.180%	---
Cube placement by the robot *	$u_{B1}$	0.020%	equal
Sensor distance sensitivity	$u_{B2}$	0.096%	equal
Illumination effect **	$u_{B3}$	0.700%	equal
Conveyor movement effect *	$u_{B4}$	0.005%	equal
Range of measured values	$u_{B5}$	1.216%	equal
Microclimate ***	$u_{B6}$	0.100%	equal

* value estimated based on the documentation, ** value estimated from experimental measurements, *** estimated value related to the workplace.

**Table 13 sensors-21-01797-t013:** Balance table of uncertainties for the brown cube.

Uncertainty Balance for the Brown Calibration Cube Moving on the Conveyor						
Measurement Impact		Standard Uncertainty	Distribution	Sensitivity Coefficient	Uncertainty Contribution	
		$c_{i}\cdot u_{i}\;\left(\%\right)$		$c_{i}$	$c_{i}\cdot u_{i}\;\left(\%\right)$	${\left(c_{i}\cdot u_{i}\right)}^{2}\;\left(\%\right)$
$u_{A}$	Repeatability	0.180	---	1	0.180	0.032400
$u_{B1}$	Cube placement by the robot	0.020	equal	1	0.020	0.000400
$u_{B2}$	Sensor distance sensitivity	0.096	equal	1	0.096	0.009216
$u_{B3}$	Illumination effect	0.700	equal	1	0.700	0.490000
$u_{B4}$	Conveyor movement effect	0.005	equal	1	0.005	0.000025
$u_{B5}$	Range of measured values	1.216	equal	1	1.216	1.478000
$u_{B6}$	Microclimate	0.100	equal	1	0.100	0.010000
					$u_{c}\;\left(\%\right)$	1.422000
					U (%)	2.844000

## Data Availability

Not applicable.

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
