# Peer review of "Intelligent Dynamic Identification Technique of Industrial Products in a Robotic Workplace"

_sensors, 2021, doi:10.3390/s21051797_

Round 1

Reviewer 1 Report

Identification of industrial products in motion based on color is a challenging problem. The authors faced this problem using calibration cubes at different distances from the camera and at different lighting conditions. Cubes are very different from each other and the identification resulted accurate even if easy due to the differences itself.
The authors do not explore the possibility of using the use of color calibration algorithms (see TPS-3d algorithm by Menesatti et al.). I suggest to explore and discuss this possibility. Sometimes the paper is quite repetitive proposing a lot of Tables.
All the information and acronyms inside a Table and Figures should be explained.

Author Response

Identification of industrial products in motion based on color is a challenging problem. The authors faced this problem using calibration cubes at different distances from the camera and at different lighting conditions. Cubes are very different from each other and the identification resulted accurate even if easy due to the differences itself.

The authors do not explore the possibility of using the use of color calibration algorithms (see TPS-3d algorithm by Menesatti et al.). I suggest to explore and discuss this possibility.

Response 1: Yes, the input data can be adjusted using sRGB color calibration algorithms before processing. The algorithms that we explored and could be used: Thin Plate Spline interpolation (TPS-3D), Partial Least Square analysis (PLS), or commercial calibration algorithm ProfileMaker (PROM). If we came to conclusion that our procedure is not sufficient for correct evaluation of color in motion, we would need to use also some of mentioned calibration method. In our case, we didn't want to add this step in advance to the process unless necessary. It is obvious that, the results of our machine learning algorithm K-nearest neighbors would be more precise. On the other side, it would increase the processing time, which is not desirable with fast dynamic scanning of products in motion.

Sometimes the paper is quite repetitive proposing a lot of Tables.

Response 2: Static and dynamic measurement tables have been joined for particular color (table 1+2 into table 1, table 3+4 into table 2 etc.) in the Results section.

All the information and acronyms inside a Table and Figures should be explained.

Response 3: Description information from tables / figures and acronyms have been added according instruction. 

Reviewer 2 Report

Abstract:

-- A technique, algorithm or model can be added in the title “Intelligent dynamic identification of industrial products in a robotic workplace”.

-- What does the type A and B refer to?

Some keywords can be deleted such as measurement, identification.

  1. Introduction

-- Have other algorithms been used in product’s identification in addition to K-nearest neighbor machine learning algorithm?

-- “according to [3, 4, 5, 6, 7, 8]” can be changed to “[3–8]”.

  1. Subject and methods

-- The unit “m” should not be italic for the Automatica.

-- Figure 1 and Figure 2 can be merged.

-- Figures 3-5 can be merged.

-- A workflow chart should be added to clearly describe the combination and processes of industrial components.

-- A table can be used to show the primary parameters of robot, color sensor and conveyor belt.

  1. Results

-- The section should be reorganized according to a logical structure.

-- The first two paragraphs should be placed in the Section 2. Subject and method.

-- Some short paragraphs can be merged.

-- Table 1 and Table 2 can be merged to comparatively show the differences between when the cube stops and in motion on the conveyor. Similarly, they can be also done for Tables 2 and 3, ……, 11 and 12.

-- Please explain the title “3.1. Statistical evaluation if measured data”.

-- The formulas of statistical methods should be placed in the Section 2. Subject and method.

A discussion section was missing. It can be a separate section or added in the 3. Results.

  1. Conclusion

-- The section should be rewritten to give some general description or future directions in the study of intelligent dynamic identification of industrial products in a robotic workplace according to the results.

Author Response

Abstract:

-- A technique, algorithm or model can be added in the title “Intelligent dynamic identification of industrial products in a robotic workplace”.

Response 1: Yes, as we describe identification technique, we have changed the title accordingly.

-- What does the type A and B refer to?

Response 2: The type A and B refers to: type A and type B standard uncertainties, which are methods of statistical analysis used for evaluation of measured data. Same has been updated in the abstract.

Some keywords can be deleted such as measurement, identification.

Response 3: We have revised keywords and updated to most relevant Keywords: production line, color sensor, uncertainties, control charts, machine learning.

  1. Introduction

-- Have other algorithms been used in product’s identification in addition to K-nearest neighbor machine learning algorithm?

Response 4: Yes, the input data can be adjusted using sRGB color calibration algorithms before processing. The algorithms that we explored and could be used: Thin Plate Spline interpolation (TPS-3D), Partial Least Square analysis (PLS), or commercial calibration algorithm ProfileMaker (PROM). If we came to conclusion that our procedure is not sufficient for correct evaluation of color in motion, we would need to use also some of mentioned calibration method. In our case, we didn't want to add this step in advance to the process unless necessary. It is obvious that, the results of our machine learning algorithm K-nearest neighbors would be more precise. On the other side, it would increase the processing time, which is not desirable with fast dynamic scanning of products in motion.

-- “according to [3, 4, 5, 6, 7, 8]” can be changed to “[3–8]”.

Response 5: Citation references have been changed according the instruction.

  1. Subject and methods

-- The unit “m” should not be italic for the Automatica.

Response 6: Unit format have been changed according the instruction.

-- Figure 1 and Figure 2 can be merged.

Response 7: Yes, figures 1 and 2 have been merged.

-- Figures 3-5 can be merged.

Response 8: Figures 4 and 5 related to sensor have been merged. 

-- A workflow chart should be added to clearly describe the combination and processes of industrial components.

Response 9: Methodology of measurement process through all combinations of industrial products and monitored factors is described in the workflow diagram and have been added into Subjects and methods section.

-- A table can be used to show the primary parameters of robot, color sensor and conveyor belt.

Response 10: Primary parameters for robot, color sensor and conveyor belt have been added into separate tables in the Subjects and methods section.

  1. Results

-- The section should be reorganized according to a logical structure.

see responses bellow

-- The first two paragraphs should be placed in the Section 2. Subject and method.

Response 11: Yes, correct. The description of sensor output values and calculation of resulting color percentage have been moved from Results into Subject and methods section.

-- Some short paragraphs can be merged.

see response bellow

-- Table 1 and Table 2 can be merged to comparatively show the differences between when the cube stops and in motion on the conveyor. Similarly, they can be also done for Tables 2 and 3, ……, 11 and 12.

Response 12: Static and dynamic measurement tables have been joined for particular color (table 1+2 into table 1, table 3+4 into table 2 etc.) in the Results section. This result into merge of some paragraphs in the Results section.

-- Please explain the title “3.1. Statistical evaluation if measured data”.

Response 13: It was a typo. Corrected: Statistical evaluation of measured data. 

-- The formulas of statistical methods should be placed in the Section 2. Subject and method.

Response 14: Yes, correct. The description of sensor output values and calculation of resulting color percentage have been moved from Results into Subject and methods section.

A discussion section was missing. It can be a separate section or added in the 3. Results.

see response bellow

  1. Conclusion

-- The section should be rewritten to give some general description or future directions in the study of intelligent dynamic identification of industrial products in a robotic workplace according to the results.

Response 15: Conclusion section has been changed. A discussion section has been created in regards to proposed cheaper alternative where simple technologies already proven in the industry operations are used in
more complex identification cases. Results and broader implications are discussed. Future research directions where we expect enhancements of identification methods by using 3D optical methods, but still with standard industrial components, have been given in the conclusion section.

Round 2

Reviewer 1 Report

The paper has been deeply implemented, but the Discussion should contain arguments with respect to the present literature on the results obtained.

Author Response

The paper has been deeply implemented, but the Discussion should contain arguments with respect to the present literature on the results obtained.

Answer: Yes, we have added corresponding literature, when discussing the obtained results.

Also, we would like to thank you for your valuable comments and helpful suggestions for our paper so far.

Reviewer 2 Report

Most of the issues raised in the first review have been addressed, however, some issues still need to be addressed as follows.

-- Lines 621-622. Please correct the sentence “If during the process occurs measured value which does not fall inside the control 621 limits of any color (at given workplace setting), ……”.

-- Graphs 1-6 can be merged to better compare the of three color component control charts.

-- The resolution of some figures (e.g., Figure 2) should be improved.

-- Some short paragraphs can be merged such as the paragraphs in lines 322-364.

-- Some references are required to approve or support the results in the 4. Discussion.

-- 5. Conclusions. Generally speaking, references do not appear in the conclusions. Additionally, the first sentence is too long. The section should be rewritten according to the primary results.

Author Response

Most of the issues raised in the first review have been addressed, however, some issues still need to be addressed as follows.

-- Lines 621-622. Please correct the sentence “If during the process occurs measured value which does not fall inside the control 621 limits of any color (at given workplace setting), ……”.

Answer: Yes, we have rephrased the sentence:

If measured value outside the control limits of any color occurs during the process, this value is tested by an algorithm trained for that setting.

-- Graphs 1-6 can be merged to better compare the of three color component control charts.

Answer: Yes, we have merged graphs for particular color, for better visibility of differences between static and dynamic measurements.

-- The resolution of some figures (e.g., Figure 2) should be improved.

Answer: Yes, we redesigned and replaced the figure in the updated paper.

-- Some short paragraphs can be merged such as the paragraphs in lines 322-364.

Answer: If I understand correctly your comment, it is related to small paragraphs. One starting at line 322, and second paragraph ending on the line 364. These we have changed.

-- Some references are required to approve or support the results in the 4. Discussion.

Answer: Yes, we have added corresponding literature, when discussing the obtained results.

-- 5. Conclusions. Generally speaking, references do not appear in the conclusions. Additionally, the first sentence is too long. The section should be rewritten according to the primary results.

Answer: Yes, conclusion have been reevaluated and now it’s available in common section – Discussion.

Also, we would like to thank you for your valuable comments and helpful suggestions for our paper so far.